# Locus coeruleus neurons encode the subjective difficulty of triggering and executing actions

**Pauline Bornert, Sebastien Bouret** *

Motivation, Brain and Behavior Team, Institut du Cerveau et de la Moelle épinière (ICM), INSERM UMRS 1127, CNRS UMR 7225, Pitié-Salpêtrière Hospital, Paris, France

* sebastien.bouret@icm-institute.org

## Abstract

The brain stem noradrenergic nucleus locus coeruleus (LC) is involved in various costly processes: arousal, stress, and attention. Recent work has pointed toward an implication in physical effort, and indirect evidence suggests that the LC could be also involved in cognitive effort. To assess the dynamic relation between LC activity, effort production, and difficulty, we recorded the activity of 193 LC single units in 5 monkeys performing 2 discounting tasks (a delay discounting task and a force discounting task), as well as a simpler target detection task where conditions were matched for difficulty and only differed in terms of sensory-motor processes. First, LC neurons displayed a transient activation both when monkeys initiated an action and when exerting force. Second, the magnitude of the activation scaled with the associated difficulty, and, potentially, the corresponding amount of effort produced, both for decision and force production. Indeed, at action initiation in both discounting tasks, LC activation increased in conditions associated with lower average engagement rate, i.e., those requiring more cognitive control to trigger the response. Decision-related activation also scaled with response time (RT), over and above task parameters, in line with the idea that it reflects the amount of resources (here time) spent on the decision process. During force production, LC activation only scaled with the amount of force produced in the force discounting task, but not in the control target detection task, where subjective difficulty was equivalent across conditions. Our data show that LC neurons dynamically track the amount of effort produced to face both cognitive and physical challenges with a subsecond precision. This works provides key insight into effort processing and the contribution of the noradrenergic system, which is affected in several pathologies where effort is impaired, including Parkinson disease and depression.

## Introduction

The locus coeruleus noradrenergic (LC/NA) system is involved in various functions such as arousal, stress, attention, and decision-making (see [1] for review). We recently proposed a role in effort processing [2–5]. Effort encompasses multiple notions, but it could be generally

**Data Availability Statement:** The data underlying this study can be found in DOI 10.17605/OSF.IO/PYVSA.

**Funding:** The present work was supported by the following institutions: Centre National de la

Recherche Scientifique (CNRS) to SB; Institut du Cerveau (ICM) to SB; Agence Nationale pour la Recherche (ANR) project NEUROEFFORT to SB and Universite Paris Sorbonne to PB. The funders had no role in study design, data collection and analysis, decision to publish, or preparation of the manuscript.

**Competing interests:** The authors have declared that no competing interests exist.

**Abbreviations:** AIC, Akaike information criteria; BIC, Bayesian information criteria; GLM, generalized linear model; LC, locus coeruleus; LC/NA, locus coeruleus noradrenergic; RT, response time; WTW, willingness to work.

defined as a mobilization of resources to face challenges [6,7]. In that sense, effort constitutes a cost and discounts reward value: All other things being equal, animals tend to minimize energy expenditure [8–15]. But for a given level of difficulty, effort production increases performance, such that it also increases reward rate. In the case of physical effort, the mobilized resources allow us to perform challenging actions. In the case of cognitive effort, the mobilized resources enhance executive control over behavior to meet task goals and inhibit automatic responses [7,16–20]. Even if the nature of these resources remains elusive, both from a theoretical and from a physiological perspective, clarifying the dynamic relation between LC activity and effort production would provide a critical insight into the mechanism of resource mobilization associated with effort production. Given the strong influence of the LC/NA system on cognition, and its potential implication in major disorders such as Parkinson disease and depression where effort is often impaired, understanding the relation between LC and the multiple aspects of effort is essential.

Besides our own work showing a strong implication of the LC/NA system in physical effort, several studies suggest that the LC could also play a critical role in cognitive aspects of effort (which can be referred to as cognitive control or mental load). Indeed, several pharmacological studies showed a clear relation between noradrenergic levels and performance in task requiring cognitive control [21,22]. The relation between LC activity and cognitive effort is also supported by the dual relation between pupil size and LC activity on one hand and cognitive effort on the other hand [5,23–27]. But to our knowledge, the relation between LC activity and cognitive effort remains very speculative.

To capture the dynamic relation between LC and effort production, both in the physical and in the cognitive domain, we compared LC activity across tasks manipulating quantitatively the amount of expected reward and/or the difficulty levels of both cognitive and physical challenges, which monkeys try to face by producing equivalent levels of effort. We hypothesized that LC neurons would be activated at the time when monkeys did produce effort to face the challenge at hand, both in the cognitive domain (triggering an action that they would spontaneously avoid) and in the physical domain (producing a high level of force). Critically, the magnitude of LC activation should scale with the subjective difficulty and with the amount of effort produced. To manipulate cognitive effort, we used discounting tasks where erroneous trials were repeated, such that monkeys would need to overcome a natural tendency to disengage when offered a low value option. Indeed, cognitive control is critical for overriding default responses [19,28–30]. Thus, we assumed that the amount of cognitive control allocated when triggering the actions increased when the value of the option decreased. To manipulate physical effort, we required monkeys to squeeze a grip with various levels of forces.

Practically, we examined LC activity around action onset in 3 tasks: (1) a delay discounting task, where conditions only differed in terms of cognitive constraints, but not in terms of sensory-motor constraints; (2) a force discounting task, which included both cognitive and sensory-motor constraints; and (3) a control target detection task, where conditions were matched in terms of difficulty but differed in terms of sensory-motor processes. We expected LC activity to increase when monkeys decided to trigger the action, with an increase in activation level when the option's value was low, i.e., when cognitive effort increased. We also expected an activation of LC neurons while monkeys exerted force on the grip, with a modulation of the activation as a function of the level of physical effort. We expected no modulation of LC activity across conditions in the control target detection task since difficulty (and therefore effort levels) were matched across conditions. The data were globally compatible with these predictions, reinforcing the idea that the LC reflects effort production with a subsecond precision, to face both physical and cognitive challenges.

## Materials and methods

### Animals

A total of 5 male rhesus macaques (*Macaca mulatta*) were included in the study: 2 in the delay discounting study (Monkey L, 9 kg and Monkey T, 9.5 kg), 1 in the target detection task study (Monkey J, 12 kg), and 2 in the force discounting study (Monkey D, 11 kg and Monkey A, 10 kg). During testing days, they received water as reward, and on nontesting days, they received amounts of water matching their physiological needs. The experimental procedures for the force discounting task and target detection task studies were designed in association with the veterinarians of the ICM Brain and Spine Institute, approved by the Regional Ethical Committee for Animal Experiment (CREEA IDF n˚3, agreement number A-75-13-19), and performed in compliance with the European Community Council Directives (86/609/EEC). The experimental procedures of the delay discounting study followed the ILAR Guide for the Care and Use of Laboratory Animals and were approved by the NIMH Animal Care and Use Committee (ASP LN 14(06)).

### Behavior

During sessions, the monkeys squatted in a primate chair, in front of a computer screen on which the visual stimuli of the task were displayed. For the force discounting task and the target detection task, force grips (M2E Unimecanique, Paris, France, pneumatic for force discounting task, electronic for target detection task) were mounted on the chair at the level of the monkey's hands: 1 for the force discounting task and 3 for the target detection task (left, right, and middle grips). For the delay discounting task, a touch sensitive bar was installed on the chair at the level of the monkey's hands. The monkeys received water reward from a tube placed between their lips but away from their teeth.

### Tasks

Behavioral paradigms were controlled using the REX system (NIH, Maryland, United States of America) and Presentation software (Neurobehavioral Systems, California, USA) for the force discounting and delay discounting tasks and using the EventIDE software (OkazoLab, London, United Kingdom) for the target detection task.

**Delay discounting task.** To examine the link between LC activity and cognitive effort, we used a task in which sensory and motor requirements were virtually equivalent across all options, to facilitate the comparison of cognitive processes across conditions. Therefore, the task involved a delay to the reward as a cost. Practically, monkeys had to release a touch sensitive bar after a go signal in order to get a reward of a certain size, after a certain delay (Fig 1A). The reward size and delay factors were orthogonalized. There were 9 experimental conditions, corresponding to the combinations of 3 levels of reward size (1, 2, or 4 drops of water) and 3 levels of delay to reward delivery (400 to 600 ms, 3,000 to 4,200 ms, and 6,000 to 8,400 ms; Fig 1B). Within sessions, these 9 conditions were randomly distributed with equal probability of appearance. Each combination of reward size/temporal delay was associated to its own visual cue, displayed at the beginning of the trial to indicate the reward–delay contingency of the trial. Each trial began when the monkey touched the bar, and the cue was then displayed for 400 ms (Fig 1A). Then, a red dot appeared, for a random duration between 1,000 and 2,000 ms, after which it turned green (go signal), indicating that the monkey had to release the bar. If the monkey responded between 200 and 1,000 ms, the green point turned blue, as a feedback to indicate correct performance. On correct trials, monkeys had to wait for the delay indicated by the cue before getting the announced liquid reward. A new trial then began after a 1,000-ms

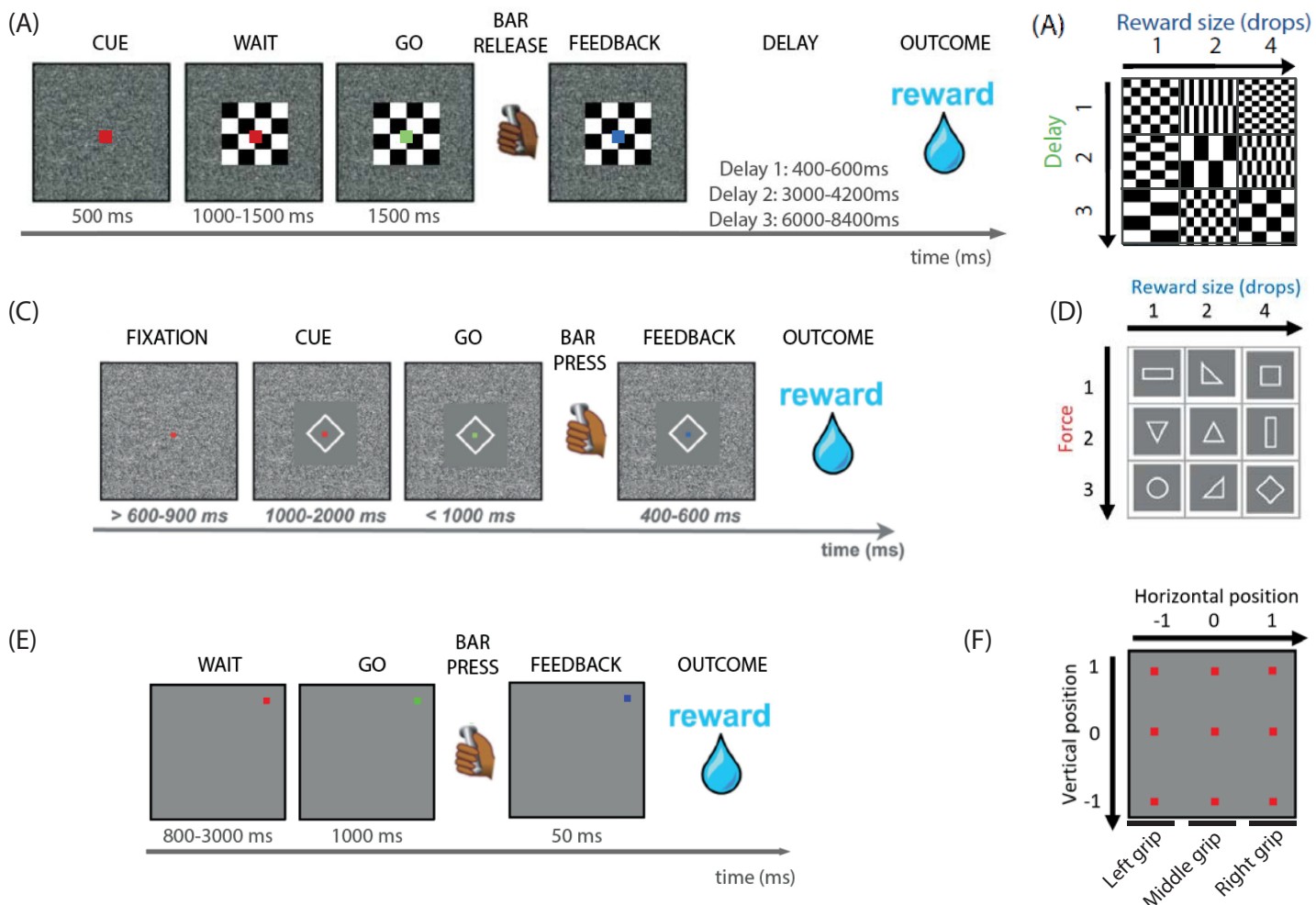

**Fig 1. Summary of the delay discounting, force discounting, and target detection tasks. (A, B)** Delay discounting task. (A) At the beginning of each trial, a cue appeared, indicating the combination of reward size and delay. After 500 ms, a red point appeared, superimposed on the cue, indicating that the monkey had to wait. After a random interval of time (1,000 to 1,500 ms), the red point turned green (go signal), for 1,500 ms. If the monkey released the bar between 200 and 1,500 ms after the onset of the green point, the point turned blue, indicating correct performance. In correct trials, the reward announced by the cue was delivered after a random delay comprised in the interval of time announced by the cue. If the monkey did not perform any bar release or performed it too early or too late, the same trial started again. (B) The task had 9 conditions, a combination of 3 levels of delay (400 to 600; 3,000 to 4,200; and 6,000 to 8,400 ms) and 3 reward sizes. Each condition was associated to a specific cue. **(C, D)** force discounting task. (C) At the beginning of each trial, a red dot appeared, and the monkey had to fixate his gaze onto it. After a random period of time, a cue appeared, indicating the combination of reward and force that was offered. The monkey had to maintain fixation on the red dot for the whole duration of cue presentation (1,000 to 2,000 ms). The red dot then turned green (go signal), and the monkey had to trigger the response within 1,000 ms, i.e., press the grip to reach the instructed force threshold. When the threshold was reached, the dot turned blue but monkeys had to maintain the force above threshold for 400 to 600 ms in order to obtain the reward predicted by the cue. (D) The task had 9 conditions, a combination of 3 levels of reward (1, 2, or 4 drops of water) and of 3 levels of force. Each combination was associated to a specific cue. **(E, F)** Target detection task. (E) Trials began with the onset of a red dot (wait signal) in 1 of 9 positions on the screen. After a random delay (800 to 3,000 ms), the red dot turned green (go signal), and the monkey had to press one of the 3 available grips within 1,000 ms. The force threshold was set just above baseline, such that any force would be sufficient. If the monkey pressed the grip, the dot turned blue and a reward (fixed amount of liquid) was delivered. In case of omission (no response) or anticipation (response during wait period), the trial was restarted. (F) The dots could appear in 9 different locations on screen, defined by 3 potential horizontal and 3 potential vertical coordinates. Horizontal coordinates indicated which grip had to be used: If the dot was on the left side of the screen, the monkey had to press the left-side grip, if the dot was on the right side of the screen, the monkey had to press the right-side grip, and if the dot was in the middle column of the screen, the monkey had to press the middle grip.

intertrial interval. If the monkey failed to release the bar during the appearance of the green point (omission), or released the bar too early (anticipation), the trial was interrupted, all visual stimuli disappeared from the screen, and the same trial started again.

**Force discounting task.** The force discounting task was described in detail in [5]. In summary, the task consisted in squeezing a pneumatic grip above a force threshold in order to get

a reward, delivered after each successful force exertion (Fig 1B). At the beginning of each trial, a red dot was presented on the screen before the onset of a visual cue. Subjects were instructed to fixate their gaze on the dot throughout the trial. The cue indicated the amount of reward at stake in the trial (3 reward levels, 1, 2, or 4 drops of water) and the minimum amount of force to exert in order to get that reward (position of the force threshold, 3 force levels; Fig 1C). The 2 dimensions were orthogonalized; therefore, the task had 9 equiprobable conditions, presented in pseudo-random order. After a variable amount of time (1,500 ± 500 ms from cue onset), the dot turned green (go signal), and the monkeys had 1,000 ms to initiate the action, i.e., to start squeezing the pneumatic grip. If the force exerted reached the minimum threshold instructed by the cue, the dot turned blue and remained blue if the force exerted remained above the threshold for 500 ± 100 ms. If the force had been maintained above threshold for the whole period, the reward announced by the cue was delivered. An error was registered if the monkey did not engage at all (no squeezing), engaged (started squeezing) before the go signal, did not press with enough force, did not maintain the force exerted above threshold for long enough (during the whole duration of the blue dot), or broke gaze fixation on the dot. Erroneous trials were repeated until completion. Note that there was no upper threshold for the force that the monkeys had to exert on the force-sensitive grip. Still, even if they could solve the task by producing a very high force on all trials, they did use the visual cues to adjust the amount of force produced to the required level (see S1 Fig).

**Target detection task.** The target detection task was designed to control for the modulation of LC activity across conditions differing more in terms of sensory-motor features than in terms of value and/or effort. The tasks consisted in pressing a grip with very little force constraint to get a fixed reward (Fig 1E). The force threshold was set just above baseline, such that any attempt to press was successful. Critically, the force threshold and the size of the reward were equivalent across all conditions, which only differed by the location of the target on the screen and the position of the grip to be pressed. A trial began with the presentation of a red dot. The dot could appear in 9 positions (Fig 1F). The position of the dot was maintained during blocks of trials of random duration (5 to 50 consecutive trials) before changing location. After a very variable amount of time (1,900 ± 1,100 ms), the red dot turned green (go signal), and the monkey had to squeeze one of the 3 grips within the next 1,000 ms. If the dot was displayed in one of the 3 positions on the left side of the screen, the monkey had to press the left grip, if the dot was displayed in one of the 3 positions on the right side of the screen, the monkey had to press the right grip, and if the dot appeared in one of the 3 central positions, the monkey had to press the middle grip (Fig 1F). If the monkey squeezed the correct grip, the dot turned blue for 50 ms (feedback), and the reward was delivered. An error was registered if the monkey squeezed any grip before the go signal or squeezed the wrong grip during go signal presentation, which virtually never happened during recording sessions.

## Surgical procedures

Surgical procedures were the same as previously described [5,31]. Briefly, the approximate location of the LC was identified using 1.5T MRI. Under general anesthesia, a sterile surgery was performed to place the head fixation post and the recording chamber, centered stereotaxically over the body of the LC with an approximately 15° angle for all monkeys.

## Electrophysiology

Electrophysiological recordings were performed using Tungsten micro-electrodes (UEW-LEHSM3PNM, FHC, Bowdoin, Maine, USA). The electrode was placed using a stereotaxic plastic grid (Crist Instruments, Hagerstown, MD, USA) with holes 1 mm apart, inserted

through a tungsten guide tube (Crist Instruments) and lowered using a hydraulic micromanipulator (Narishige, Tokyo, Japan). LC neurons were identified using classical electrophysiological criteria [32]: a low rate of spontaneous activity (below 4 Hz), broad waveforms (>0.6 ms for the initial peak), and a modulation of firing rate across states of vigilance (evaluated by monitoring eye opening versus closing periods, global locomotor activity, and reactivity to surrounding sounds). LC neurons also display a characteristic activation pause response to brief alerting auditory or tactile stimuli (e.g., handclap), such that behavioral habituation predicted a decrease in LC responses to these stimuli. Finally, we also performed clonidine tests on a series of representative LC units at locations where LC neurons could reliably be obtained during the course of the experiments. For this, we injected the α2-receptor agonist clonidine (20 μg/kg, IM), and all identified LC units displayed a reversible decrease in firing rate (often close to zero [31,32]). None of the units identified as non-LC cells (e.g., neurons from the neighboring Mes5 nucleus, which respond to jaw movement) changed their activity after clonidine injection. A total of 92 neurons were recorded in the force discounting task ($n = 63$ in Monkey D and $n = 29$ in Monkey A), 75 in the delay discounting task ($n = 52$ in Monkey T and $n = 23$ in Monkey L), and 26 in the target detection task (Monkey J).

## Data analysis

Analyses of the data from the force discounting and target detection tasks were performed using the MATLAB software (MATLAB 2019a, Mathworks), and analyses of the data from the delay discounting task were performed on the R software (R 3.5.0, R Foundation for Statistical Computing).

We only analyzed trials in which the monkeys engaged after the go signal, i.e., tried to perform the action, in order to be able to compute response time (RT). In the delay discounting and the target detection tasks, engaging after the go signal ensured succeeding due to the simplicity of the action to be performed. In the force discounting task and the delay discounting task, preliminary inspection of the neuronal activity did not reveal differences between monkeys, so the data were pooled by task.

Regarding behavior, analyses were performed on 2 parameters: willingness to work (WTW), a binary variable (whether the monkey accepted to perform the required action or not), and RTs, a continuous variable (the time taken by the monkey to respond to the go signal and initiate the required action; see determination below). We also considered the maximum force exerted by the monkeys on the force grips during the presses as a parameter in our models (see determination below). Regarding neuronal activity, we computed spike counts around action onset and examined both changes from baseline and parametric modulations across conditions (defined by task parameters and/or behavioral variables). All the following analyses were performed after z-scoring each of the nonbinary parameters (reward, delay, force category, RT, and maximum exerted force) and spike count by neuron.

**Response time determination (target detection and force discounting tasks).** The force signal was digitized and sampled at 1 kHz. During data acquisition, RTs were evaluated online and defined as the time when the force signal reached the target threshold. But to avoid the potential influence of the threshold position, RTs were reevaluated offline and defined as the time when the force signal started to diverge from baseline, i.e., the beginning of the squeeze on the grip. In short, the mean slope of the force signal was computed in successive 10-ms windows (difference in signal value between the end and the beginning of the time window). We considered that the animal had started pressing the grip when the slope was above the maximum slope that could be detected in the baseline period outside of periods of force exertion, in 3 consecutive time windows.

**Maximum exerted force determination (target detection and force discounting tasks).**
To assess the trial-by-trial maximum exerted force in the 2 force tasks, we first subtracted the
baseline (mean of the signal in a [−500; 0 ms] epoch before action onset) from the force signal.
We then looked for the maximum in the force signal in a [0; 900 ms] epoch after action onset
(based on the RT previously determined), since signal inspection revealed that the presses
lasted between 600 and 900 ms.

**Behavioral analyses pooling sessions by monkey.**   To analyze the behavior (RT and
engagement) per monkey per task, we pooled the data from all the sessions for each monkey.
Regarding engagement, in the force discounting and delay discounting tasks in which we
expected some linear modulations of engagement by task parameters, we fit logistic regres-
sions for engagement with reward and delay in the delay discounting task and reward and
force in the force discounting task:

Delay discounting: $e \sim \text{logit}(c + \beta_R . R + \beta_D . D)$

Force discounting: $e \sim \text{logit}(c + \beta_R . R + \beta_F . F)$,

with e the binary variable engagement (0 if the monkey did not engage, 1 if the monkey did
engage), c a constant term, R the reward (z-scored), D the delay (z-scored), F the force category
(z-scored), and the β the associated coefficients.

We tried fitting the same models with a random term for the session number, but this did
not affect the size of the coefficients for the task parameters (force or delay and reward). Con-
sequently, for the sake of simplicity, we report results without these terms. Interaction terms
were not significant, and we therefore did not include them in our models.

In the target detection task, since we did not expect any linear effects of the position of the
dot, we compared engagement rates (percentage of engaged trials) across the 9 conditions
using a 1-way ANOVA with dot position as parameter (9 arbitrary levels).

Regarding analyses of RTs, we fit generalized linear models (GLMs) for RT with force cate-
gory (force discounting task) or delay (delay discounting task) and reward as parameters:

Delay discounting: $RT \sim c + \beta_R . R + \beta_D . D$

Force discounting: $RT \sim c + \beta_R . R + \beta_F . F$,

with c a constant term, R the reward (z-scored), D the delay (z-scored), F the force category (z-
scored), and the β the associated coefficients.

In order to conserve across-session variations in RT, RTs were not z-scored by session
before pooling the data of all the sessions together. However, results were similar when z-scor-
ing RTs per session before pooling the sessions. Once again, we tried fitting these models with
random terms for session numbers, which did not modify the effects of force category or delay
and reward. Thus, we report results of the GLMs without these terms. Interaction terms were
not significant and we therefore did not include them in our models.

**Sliding window procedures for effects of task parameters and behavioral indicators
(delay discounting and force discounting tasks).**   Spikes were counted in a 200-ms window
of time, which were moved forward in 25-ms steps, in a [−800; 800 ms] epoch around action
onset. In order to quantify the effects of cost (force or delay), reward and RT around action
onset on spike count, and qualify their dynamics, we then performed, for each 200 ms epoch
of the sliding window counting of spikes, a neuron-by-neuron GLM for spike count with trial-
by-trial reward, cost (force or delay) and RT as parameters.

**Model comparison for preaction spike count (delay discounting and force discounting
tasks).**   To justify including RT in the GLM explaining the preaction spike count of LC neu-
rons, we performed a model comparison procedure. Practically, we tested the hypothesis that

adding RTs to a model with only task parameters significantly improved the fit. We pooled all the trials from all the neurons, per task ($n = 75$ for delay discounting task and $n = 92$ for force discounting task). To take into account differences in baseline firing rate, we did not z-score the spike count per neuron before pooling the trials of the different neurons (but results were similar when z-scoring rate per neuron before pooling the neurons). We also did not z-score RTs per sessions to conserve across-session variations in RT, but similar results were obtained when z-scoring RTs per session before pooling the sessions. After pooling the sessions, we fit 3 different GLMs to the spike count of the neurons (using the fitglm function in MATLAB and the bic.glm function in R): 1 with only RT as regressor (1), 1 with only task parameters as regressors (2), and 1 with all task parameters and RT (3):

Delay discounting:

1. $Spk \sim c + \beta_{RT} . RT$

2. $Spk \sim c + \beta_R . R + \beta_D . D$

3. $Spk \sim c + \beta_R . R + \beta_D . D + \beta_{RT} . RT$

Force discounting:

1. $Spk \sim c + \beta_{RT} . RT$

2. $Spk \sim c + \beta_R . R + \beta_F . F$

3. $Spk \sim c + \beta_R . R + \beta_F . F + \beta_{RT} . RT$,

with Spk the spike count in the window of interest (z-scored across neurons), c a constant term, R the reward (z-scored across), D the delay (z-scored), F the force category (z-scored), and the $\beta$ the associated coefficients.

We used Bayesian information criteria (BIC) to compare the fit of the models, but similar results were obtained using Akaike information criteria (AIC). Since BIC is calculated using a log scale, a difference in 1 unit indicates that the model with the lowest BIC is 10 times better than the other one, and it is commonly accepted that a model is significantly better than the alternative if the BIC difference between the 2 models is greater than 3 [4].

## Results

We will first present behavioral data that provide information about effort production across the 3 tasks and then describe the dynamic modulation of LC activity across these 3 tasks.

### Behavior

We compared behavior in the 3 tasks, all involving a manual response to a visual stimulus (go signal, a red point turning green). In each of these tasks, monkeys completed multiple trials across which actions varied in terms of sensory-motor requirements and/or in terms of reward contingencies. At the beginning of each trial, monkeys received information about the current task condition using a specific visual cue and had to adjust their behavior accordingly. We measured 2 behavioral responses: RT (the interval between the go signal and action onset) and engagement (whether they attempted to perform the trial or not). Thus, in each condition, the proportion of trials in which monkeys engaged (engagement rate) reflected their average WTW given the expected sensory-motor constraints and reward contingencies. Critically, since trials were repeated until correct completion, there was no instrumental interest in refusing to perform any trial. In such task, the optimal behavior would be to perform all trials regardless of the associated costs and benefits, because refusing to engage only increased the

delay until reward delivery. But still, monkeys failed to initiate the action more often in trials associated with higher cost and/or lower rewards, which indicates that they could not repress a natural tendency to disengage in such conditions, even if in that situation if was counterproductive. Thus, we assumed that in task conditions associated with lower WTW, engaging in the task and performing the action required a higher level of cognitive control to overcome the stronger tendency to disengage, compared to conditions in which average WTW was higher. Even if the absence of engagement could also be taken as a passive process (a lack of motivation, rather than an urge to disengage), it would still require cognitive control to override that suboptimal behavior. Hence, we used the contrast in WTW across conditions to evaluate the influence of these conditions on cognitive difficulty and, potentially, on the amount of cognitive effort mobilized to overcome that difficulty. Since all these tasks involved triggering a response to a visual target, we could also measure RT, i.e., how quickly the animal responded to the stimulus. In line with previous studies, we assumed that RTs could be affected both by sensory effects (how difficult it was to identify the target stimulus), motor effects (how difficult it was to execute the action), and cognitive effects, namely the amount of cognitive control required to trigger the action in the current condition (here, as a function of the natural tendency to disengage from the task in the current condition) [33]. Critically, the increase in RT in conditions associated with greater cognitive control could reflect both the difficulty itself and/or the mobilization of resources (i.e., time) in order to overcome that difficulty [7,34,35]. To capture the potential influence of sensory-motor and cognitive effects on RT, we compared behavior across 3 tasks manipulating both sensory-motor and reward parameters across conditions (Fig 1).

The different features of the 3 tasks are summarized in Table 1, together with the predicted influence of sensory-motor and cognitive effects on RT. In short, all 3 tasks involved detecting a simple visual target (a red dot turning green). In the delay discounting and in the force discounting task, the target stimulus was always presented in the middle of the screen and only one responding device was available. In the target detection task, the target stimulus could be presented in 1 of 9 possible positions on the screen, and there were 3 response devices (left, middle and right). In the delay discounting task, the action was a simple bar release, whereas in force discounting and target detection task, monkeys had to squeeze a grip and exert a given level of force. In the force discounting task, the level of force necessary to complete the trial was varied systematically across trials according to 3 difficulty levels, whereas in the target detection task, the level of force required was set to a minimum and equivalent across all task conditions. Finally, we also manipulated reward parameters: In both the delay discounting and force discounting tasks, we systematically varied the size of the reward (volume of juice) according to 3 levels. In the delay discounting task, we also systematically varied the delay between correct responses and reward delivery, according to 3 levels.

**Table 1. Summary of the characteristics of the 3 tasks used in this study.**

|  | Delay discounting | Force discounting | Target detection |
|---|---|---|---|
| Target detection | ✓ | ✓ | ✓ |
| Var. position (target and response) | 0 | 0 | ✓ |
| Force production | 0 | ✓ | ✓ |
| Force constraint | 0 | ✓ | 0 |
| Reward delay | ✓ | 0 | 0 |
| Reward size | ✓ | ✓ | 0 |
| **Sensory-motor constraints:** | 0 | ✓ | ✓ |
| Cognitive constraints | ✓ | ✓ | 0 |

Based on these features, we predicted that, across task conditions, WTW should differ only in delay discounting and force discounting tasks, but not in the target detection task. Accordingly, RTs should be affected only by cognitive effects in the delay discounting task. In the target detection task, however, RTs should only be affected by sensory-motor effects. Finally, in the force discounting task RTs should display both sensory-motor and cognitive effects. To test these predictions, we compared engagement rates and RT modulations across conditions in the 3 tasks.

In the delay discounting task, there were clear differences in WTW across conditions: Engagement rates showed a significant positive modulation by reward and a significant negative modulation by delay in both monkeys. For each monkey, we fit a logistic regression for WTW with reward and delay as parameters. The reward effect was significantly positive for both monkeys (Monkey T: beta = 0.40; $p < 10\text{-}11$; Monkey L: beta = 0.23; $p < 10\text{-}4$), and the delay effect was significantly negative for both monkeys (Monkey T: beta = −0.42; $p < 10\text{-}14$; Monkey L: beta = −0.31; $p < 10\text{-}8$; Fig 2A and 2B). Since in this task sensory-motor constraints were equivalent across conditions, we expected task conditions to affect RTs through cognitive effects and thus according to their influence on WTW. Indeed, in both monkeys, task parameters had significant opposite effects on RT, as measured using linear modeling (GLM; Fig 2C and 2D). Reward had a negative effect on RT (Monkey T: beta = −0.12; $p < 10^{-34}$; Monkey L: beta = −0.23; $p < 10^{-33}$), and delay had a positive effect (Monkey T: beta = 0.22; $p < 10^{-95}$; Monkey L: beta =0.27; $p < 10^{-52}$). Thus, task parameters clearly affected WTW and consequently the cognitive difficulty associated with engaging in the task and triggering the action. The corresponding differences in RTs across conditions confirmed this interpretation in terms of cognitive difficulty to trigger the actions, but also in terms of resources (time, at least) invested in order to overcome the difficulty and perform the action.

In the target detection task, sensory-motor constraints did vary across task conditions (positions of targets and of response levers), but they did not affect WTW (1-way ANOVA on session-by-session rates of engagement in each condition, dot position as parameter, $p \gg 0.05$; Fig 2E). By contrast, these sensory-motor constraints did affect RTs: We fit a 2-way ANOVA for RT with the horizontal position of the dot (i.e., position of the grip to be used: left, middle, and right) and vertical position of the dot as parameters (low, intermediate, and high). The effect of the grip used was significant (F [2] = 27.06; $p < 10^{-4}$; Fig 2F), with lower RTs when using the middle grip (post hoc $t$ test with correction for multiple comparison, $p < 0.05$), but no effect of the vertical position of the dot ($p \gg 0.05$). These constraints were also associated with differences in terms of force production: we fitted an ANOVA for the exerted force with the side of the grip as a parameter. The effect of side was significant (F [2] = 858.6; $p < 10^{-20}$) with the force applied on the middle grip being the highest (multiple comparison of means, $p < 0.01$; Fig 2G). We finally confirmed the link between RT and exerted force by looking at the relationship between RT and the peak of exerted force by fitting RT and exerted force with a linear model. The effect of the exerted force on RT was significant and negative ($p < 10^{14}$; Fig 2H). In summary, in the target detection task, the absence of WTW contrast across conditions indicated that the amount of cognitive control required to trigger the action (press) was probably equivalent across conditions. Additionally, RTs were clearly modulated by sensory-motor constraints, but there is no reason to interpret differences in RT in terms of cognitive difficulty and/or associated cognitive effort.

In the force discounting task, as described in the original paper [5], WTW was strongly modulated across conditions. Using a logistic regression, we found that engagement was modulated positively by reward and negatively by force category ($p < 0.01$ for both parameters for both monkeys; Fig 2I and 2J). Thus, as in the delay discounting task, the amount of cognitive control required to perform the task should be modulated across task conditions, and RTs

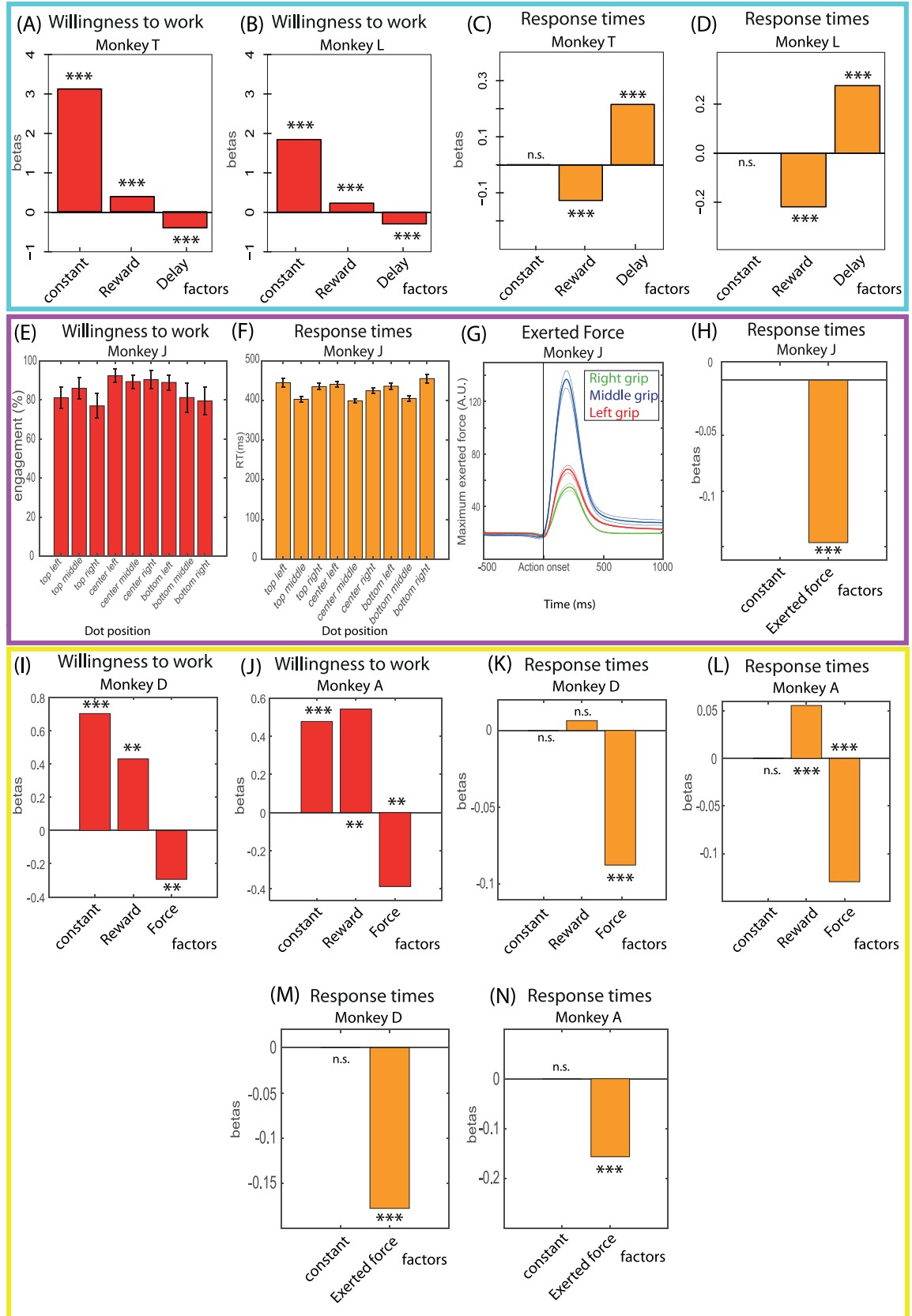

**Fig 2. Behavior in the delay discounting task, the target detection task, and the force discounting task. (A–D)** Behavior in the delay discounting task. (A, B) Coefficients of the logistic regression for engagement in trials (0 if the monkey made no action, 1 if he performed an action) with task parameters (reward and delay) as regressors, by monkey. For both monkeys, reward had a positive effect, and delay had a negative effect on the probability to engage in trials. (C, D) Coefficients of the GLM for RT with task parameters (reward and delay) as regressors, by monkey. For both monkeys, reward had a negative effect, and delay had a positive effect. **(E–H)** Behavior in the target detection task. (E) Mean engagement rate across sessions by dot position on screen. There was no difference in engagement rate across the 9 conditions (ANOVA). (F) Mean RT across sessions by dot position on screen. A 2 way-ANOVA with vertical and horizontal dot coordinates showed that RTs were significantly shorter for middle grip presses. (G) Time course of the exerted force on each grip across sessions after action onset. Thick lines represent the mean exerted force and the thinner lines represent one the SEM above and below the mean. The maximum exerted force was higher for middle grip presses (blue) than for right grip (green) or left grip (red) presses. (H) Coefficients for the GLM for RT with only the maximum exerted force as parameter. RT was longer if the maximum exerted force would be stronger. **(I–N)** Behavior in the force discounting task. (I, J) Coefficients of the logistic regression for engagement in trials (0 if the monkey made no action, 1 if he performed an action) with task parameters (reward and force) as regressors, by monkey. For both monkeys, reward had a positive effect, and force had a negative effect on the probability to engage in trials. (K, L) Coefficients for the GLM for RT with task parameters (reward and force) as regressors, by monkey. For both monkeys, reward had a positive effect (only a tendency for Monkey D), and force had a negative effect. (M, N) Coefficients for the GLM for RT with only the maximum exerted force as parameter. For both monkeys, there was a strong negative relationship between RT and the maximum exerted force. **: $p < 0.01$; ***: $p < 0.0001$; n.s., nonsignificant; error bars represent SEM. Underlying data in 10.17605/OSF.IO/PYVSA. GLM, generalized linear model; RT, response time; SEM, standard error of the mean.

should be influenced by cognitive effects. We then examined the modulations of RTs across task conditions by fitting a GLM for RTs, with the offered reward and the category of force requested as parameters. In both monkeys, RTs were positively modulated by reward (Monkey A: beta = 0.58; $p < 10^{-4}$; Monkey D: beta = 0.00076; $p = 0.47$) and negatively modulated by force category (Monkey A: beta = $-0.13$; $p < 10^{-20}$; Monkey D: beta = $-0.087$, $p < 10^{-15}$), which is the opposite of the pattern expected for pure cognitive effects, given the influence of task parameters on WTW (Fig 2K and 2L). But these effects of task parameters could also be accounted for in terms of simple motor constraints, rather than cognitive constraints. To explore that possibility, we examined the relationship between trial-by-trial RTs and exerted force (maximum exerted force on the grip during press). As was the case in the target detection task, there was a significant negative relation between RT and the amount of exerted force (GLM, $p < 0.01$ for both monkeys; Fig 2M and 2N). Thus, in this task, RTs were clearly affected by sensory-motor constraints. However, even if the pattern of WTW suggests that they might also be affected by cognitive constraints, we could not find evidence for it.

In short, even if all 3 tasks involved triggering an action in response to a visual target, the 3 tasks differed in the relative weight of sensory-motor versus cognitive constraints on RT. In the delay discounting task, responses were mostly influenced by cognitive constraints, with virtually no difference in sensory-motor processes across conditions. By contrast, in the target detection task, conditions clearly differed in terms of sensory-motor constraint but not in terms of value. In the force discounting task, the difference in behavior across conditions suggests that it involved both sensory-motor and cognitive constraints, but the relative weight of the 2 remains difficult to evaluate.

## Neurophysiology

We recorded single LC units based on previously described methodologies (see Materials and methods for details). In the delay discounting task, we recorded the activity of 75 single units from the LC ($n = 52$ in Monkey T and $n = 23$ in Monkey L). In the force discounting task, we recorded 92 LC units (in 2 monkeys, $n = 63$ in Monkey D and $n = 29$ in Monkey A). In the target detection task, we recorded 26 neurons (in 1 monkey, Monkey J). The baseline firing rate of all the recorded neurons remained relatively constant during the course of the experiments, besides the periods when monkeys were apparently asleep (eyes closed and motionless), and LC neurons were virtually silent. Within tasks, inspection of the data did not indicate any

difference between neurons recorded from the different monkeys, so the neurons were pooled by task.

**Evoked responses to action onset and action execution.** As classically described in previous studies, LC neurons were activated just before action onset [31,36–38]. For each neuron, we measured firing rates in a [−250; 0 ms] epoch before action onset and also in a baseline period, just before the onset of the go signal ([250; 0 ms] from go signal). We compared these rates using paired *t* tests and then used a second-order test (*t* tests on T values) to assess the coherence of the rate changes across the whole population in a given task. Rate significantly differed from baseline for 62/75 neurons in the delay discounting task (all increased; Fig 3A), 51/92 neurons in the force discounting task (all increased; Fig 3B); and 16/26 neurons in the target detection task (8 decreased and 8 increased; Fig 3C). At the population level, the

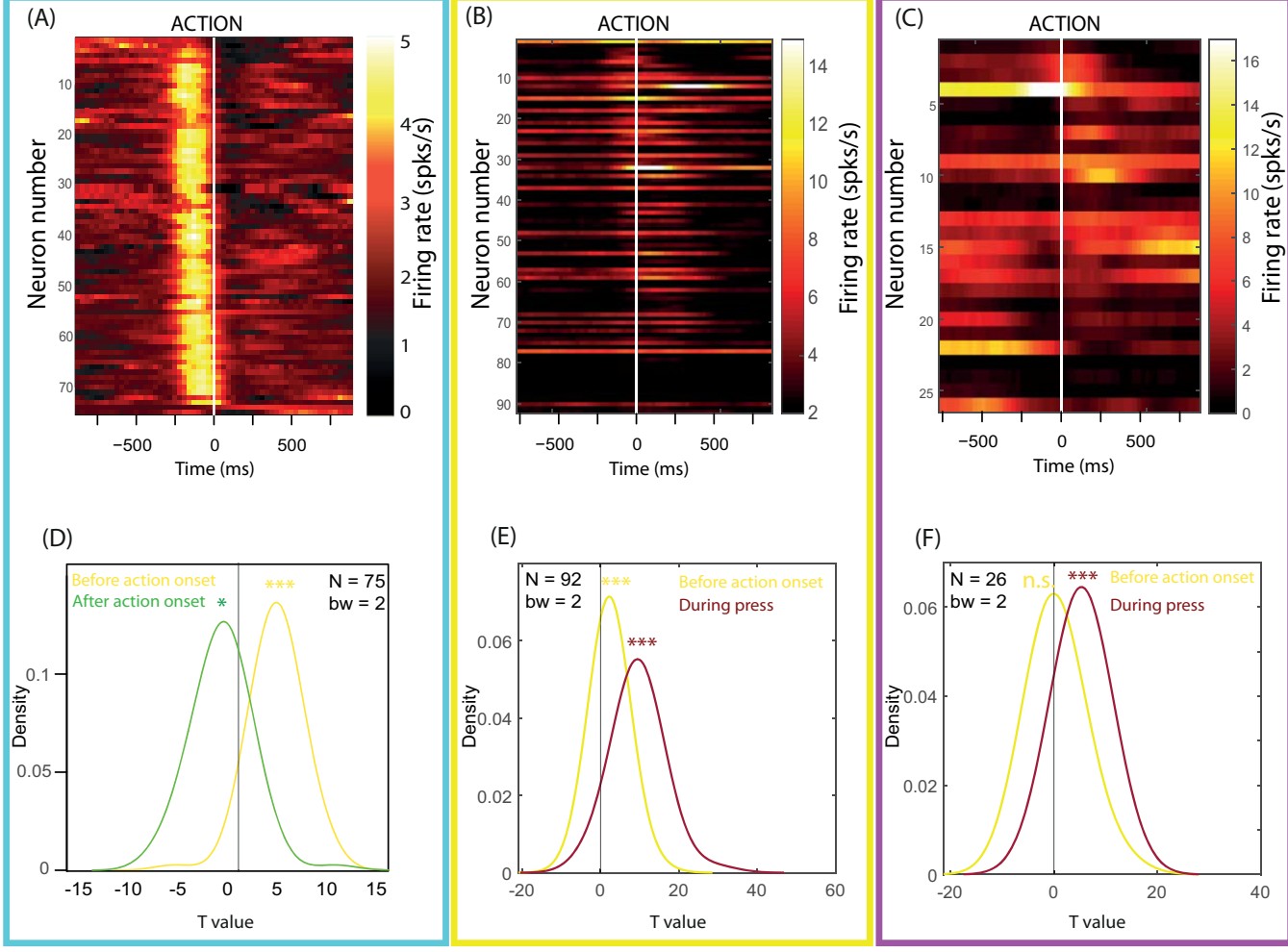

**Fig 3. Evoked responses to action onset in the delay discounting, force discounting, and target detection task. (A–C)** Mean rate (colors) of each of the recorded units (y-axis) around the time of action onset (x-axis) in (A) the delay discounting task, (B) the force discounting task, and (C) the signal-detection task. Clearer colors represent higher rates. **(D–F)** Results of the first-order and second-order *t* tests on rate in (D) the delay discounting task, (E) the force discounting task, and (F) the target detection task. Density function of the T-values of the neuron-by-neuron *t* tests comparing rate before action onset ([−250; 0 ms] from action onset, yellow), during action execution ([0; 600 ms] from action onset, dark red) or after action onset ([0; 250 ms] after action onset, green) to baseline rate (rate before go signal, [−250; 0 ms]). (D) In the delay discounting task, across the population, rate increased before action onset and decreased after action onset. (E) In the force discounting task, across the population, rate was above baseline before action onset and during action execution. (F) In the target detection task, across the population, rate was not different from baseline before action onset and was higher than baseline during action execution. *: *p* < 0.05; ***: *p* < 0.001; n.s., nonsignificant. Underlying data in 10.17605/OSF.IO/PYVSA.

activation only reached significance in the delay and force discounting tasks (second level analysis: delay discounting: (t(74) = 9.56; $p < 0.001$; Fig 3D; force discounting: t(91) = 8.88; $p < 10^{-13}$, Fig 3E). In the target detection task, the change in rate did not reach significance at the population level (t(25) = 0.57; $p = 0.58$; Fig 3F).

In addition to the activation right before action onset, LC neurons were activated during the action itself in both tasks involving force production (force discounting and target detection). In the force discounting task, in the [0; 600 ms] epoch after action onset, corresponding to the time during which the force was exerted (S1 Fig), 50/92 neurons showed a significant change in rate compared to pre-go signal rate (10 decreased and 40 increased; Fig 3B), and the activation was clearly significant at the population level (t(91) = 15.29; $p < 10^{-26}$; Fig 3E). In the target detection task, in the [0; 500 ms] after action onset during which force was exerted (Fig 2G), 20/26 neurons showed a significant change in rate during force production (6 decreased and 14 increased), and the activation was significant at the population level (t(25) = 7.38; $p < 10^{-7}$; Fig 3F). In the delay discounting task, the action was much shorter and less demanding (a small hand movement). In that task, firing rate decreased after action onset, compared to the baseline rate. The change was significant for 36/75 neurons (all decreased, t (74) = −2.14; $p = 0.035$; Fig 3A and 3D).

In short, LC neurons responded both before action onset and during force production. The activation before action onset was especially pronounced in discounting tasks, where triggering the action could involve some form of cognitive effort. LC neurons were also activated during force production, potentially in relation with the corresponding physical effort. To further assess the relation between LC activation and instantaneous effort production, we quantified the modulations of firing rate across task conditions in each of the 3 tasks.

## LC and cognitive effort: Action triggering–related activity

Since the activation of LC neurons before action onset seemed to be associated with the triggering of the action and since triggering that action seemed to be more cognitively challenging (less spontaneous) when the associated outcome value decreased, we examined the modulation of LC activity just before action onset across task conditions in each of the 3 tasks. We also examined the influence of RTs, which, in absence of sensory-motor constraints, capture trial-by-trial modulations of cognitive difficulty and/or the associated cognitive effort to trigger the response (see above, "Behavior").

**Delay discounting task.** To examine the influence of task parameters on action triggering–related activity in the delay discounting task in each neuron, we fit spike counts right before action onset with a GLM using reward and delay as parameters. The activity of an example unit encoding task parameters is shown in Fig 4A and 4B. A total of 9/75 LC units significantly encoded reward, negatively. Even if that number was close to the number of neurons expected by chance ($n = 10$) given a population of that size ($n = 75$), the negative effect of reward was significant across the population (second-order $t$ test: t(74) = −4.18; $p < 0.001$; Fig 4C). There was also a significant positive effect of delay for 20/75 neurons, and this positive effect of delay was consistent across the population (t(75) = 5.44; $p < 0.001$; Fig 4C). Thus, when monkeys triggered the action, the direction of the effects of task parameters on LC activity was opposite to what we reported for WTW (Fig 2I and 2J).

Next, we examined the influence of RT on action triggering–related activity by fitting a GLM. The activity of an example unit encoding RT positively is shown in Fig 4D. The firing of 12/75 units exhibited a significant effect of RT (positive for 11 of them), and the positive influence of RT was significant across the population (second-order $t$ test: t(74) = 4.18; $p < 0.001$; Fig 4E). When task parameters were included as covariates of RT in the regression model, 8/75

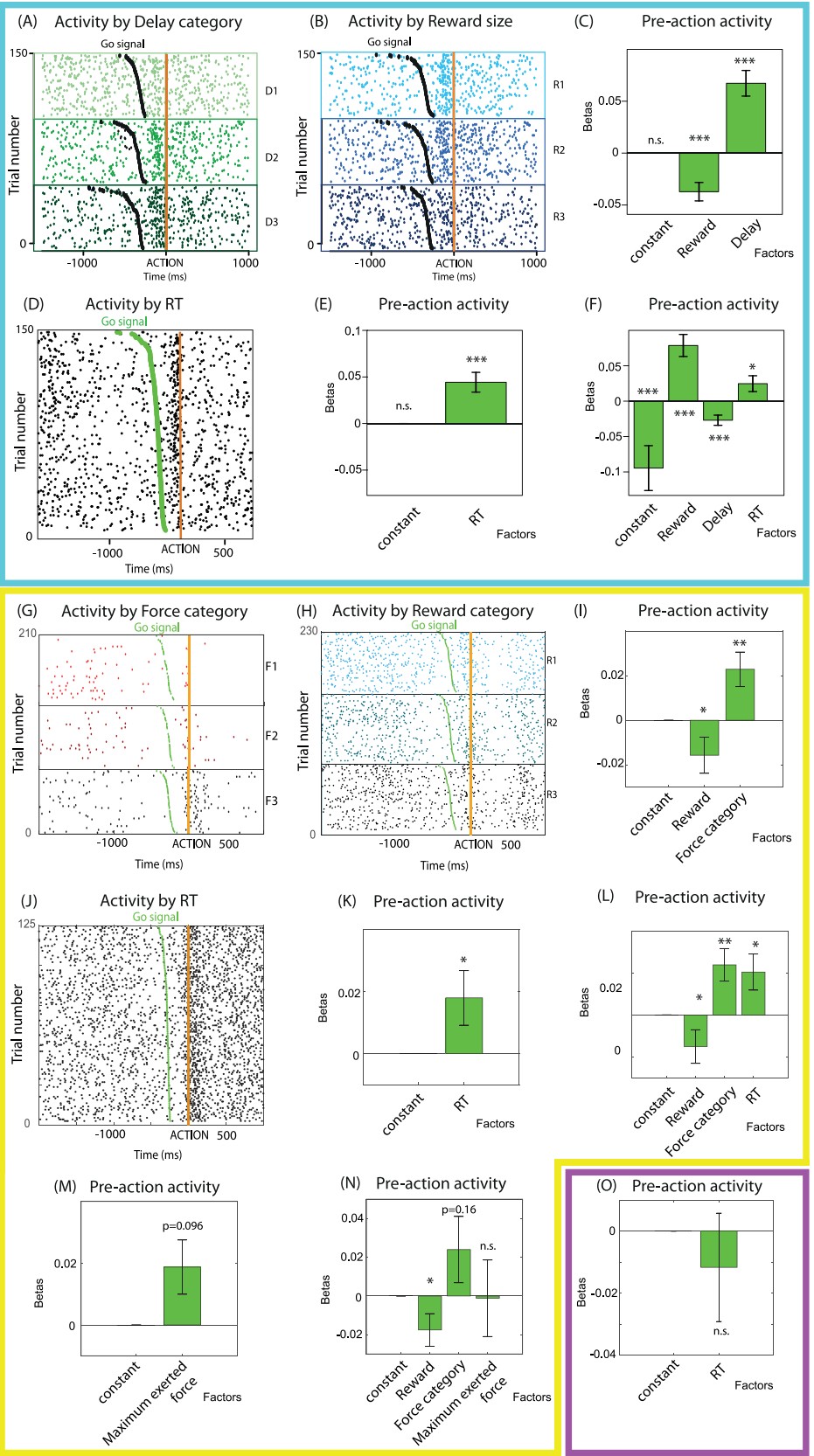

**Fig 4. Encoding of proxies for cognitive effort to trigger the action and physical effort to execute it, before action onset in (A–F) the delay discounting task, (G–M) the force discounting task, and (N) the target detection task.** (A, B) Raster plots of the activity of 2 representative LC units encoding task parameters (delay, (A) and reward, (B)) recorded in the delay discounting task, around action onset (orange vertical line). Green dots represent the go signal. Trials are split by (A) delay levels (D1,2,3) or (B) reward size (R1,2,3). Unit (A) encoded delay positively before action onset, i.e., was more active before action onset in high-delay conditions. Unit (B) encoded reward negatively, i.e., was less active before action onset in high-reward conditions. (C) Summary (mean and SEM) of the coefficients of the neuron-by-neuron GLMs for spike count before action onset with reward and delay as parameters, in the delay discounting task. Across the population, neurons encoded reward negatively and delay positively. (D) Raster plot of the activity of a representative LC unit encoding RT positively around action onset (orange line) in the delay discounting task. This unit was more active when RT (interval between go signal, green dot, and action onset) was longer. (E, F) Summary (mean and SEM) of the coefficients of the neuron-by-neuron GLMs for spike count before action onset with (E) RT alone or (F) RT, reward, and delay as parameters in the delay discounting task. (E) Across the population, neurons encoded RT positively, (F) and this positive encoding was true over and above the encoding of reward (negatively) and delay (positively). (G-H) Raster plots of the activity of 2 representative LC units encoding task parameters (force category, (G) and reward, (H)) recorded in the force discounting task, around action onset (orange vertical line). Green dots represent the go signal. Trials are split by (G) force levels (F1,2,3) or (H) reward size (R1,2,3). Unit (G) encoded force positively before action onset, i.e., was more active before action onset in high-force conditions. Unit (H) encoded reward negatively, i.e., was less active before action onset in high-reward conditions. (I) Summary (mean and SEM) of the coefficients of the neuron-by-neuron GLMs for spike count before action onset with reward and force category as parameters, in the force discounting task. Across the population, neurons encoded reward negatively and force category positively. (J) Raster plot of the activity of a representative LC unit encoding RT positively around action onset (orange line) in the force discounting task. This unit was more active when RT (interval between go signal, green dot, and action onset) was longer. (K, L) Summary (mean and SEM) of the coefficients of the neuron-by-neuron GLMs for spike count before action onset with (K) RT alone or (L) RT, reward, and force category as parameters, in the force discounting task. (K) Across the population, neurons encoded RT positively, (L) and this positive encoding was true over and above the encoding of reward (negatively) and force category (positively). (M, N) Summary (mean and SEM) of the coefficients of the neuron-by-neuron GLMs for spike count before action onset with (M) the maximum exerted force on the grip alone or (N) the maximum exerted force on the grip, reward and force category as parameters, in the force discounting task. (M) Across the population, neurons marginally encoded the maximum exerted force, (N) but this effect was not present over and above effects of task parameters (force category and reward). (O) Summary (mean and SEM) of the coefficients of the neuron-by-neuron GLMs for spike count before action onset, with RT as parameter, in the target detection task. Across the population, neurons did not encode RT. *: $p < 0.05$; **: $p < 0.01$; ***: $p < 0.001$; n.s., nonsignificant. Underlying data in 10.17605/OSF.IO/PYVSA. D, delay; F, force; GLM, generalized linear model; LC, locus coeruleus; R, reward; RT, response time; SEM, standard error of the mean.

neurons still exhibited a significant positive effect of RT. Critically, the positive influence of RT was still significant at the population level (t(74 = 2.21; $p = 0.024$; Fig 4F). Note that both reward and delay still had significant and opposite effects on LC activity (reward, significant for 8/75 neurons, 7 negative and 1 positive, second-order test: t(74) = −3.74, $p = 0.0003$; delay: significant for 17/75 neurons, all positive, second-order test: t(74) =5.05, $p < 10^{-5}$).

Finally, we compared the fit of the GLMs for spike count including RT alone, task parameters, and a full model with both RT and task parameters. We fit each of these 3 GLMs on the entire data set (all trials of all sessions of both monkeys, spike count not normalized by neuron) and compared the BIC of those 3 models. The BIC of the model with reward, delay, and RT was lowest (BIC = 35,742), compared to the BIC of the model with only RT (BIC = 35,874) and to the BIC with only task parameters (BIC = 35,760). Since the difference in BIC between the best model (RT and task parameters) and the others was greater than 5, we considered it to be significantly better than the alternative models. Thus, modulation of LC activity across conditions could be accounted for by both task parameters, which are related to the difficulty to make the decision (to perform that action), and to the RT, which increases with the difficulty to trigger the action and reflects the amount of resources (at least time) invested in triggering the action.

**Force discounting task.** We assessed the encoding of task parameters by LC neurons using a GLM for spike count before action onset with reward and force category as parameters. The activity of example LC units encoding task parameters is shown in Fig 4G and 4H. The

effect of reward was only significant for 6/92 neurons (1 positive and 5 negative), which is less than the number of neurons expected by chance ($n = 13$) for a sample of 92 neurons. At the population level, however, the effect of reward was significantly negative (second-order $t$ test: t (91) = −2.05; $p = 0.04$; Fig 4I). Similarly, the effect of force was only significant for 7/92 neurons (1 negative and 6 positive), but it was significantly positive at the level of the population (t (91) = 2.72; $p = 0.0079$; Fig 4I). Thus, the influence of task parameters on LC activity at time of action triggering mirrors their influence on WTW.

We examined the encoding of RT by LC neurons by fitting a GLM for spike count with RT alone (in addition to the constant term). The encoding of RT by an example unit is shown in Fig 4J. The effect of RT was only significant for 10/92 neurons (4 negative and 6 positive), but it was consistently positive at the level of the population (t(91) = 2.01; $p = 0.048$; Fig 4J). To evaluate the possibility that the relation between spike counts and RT was confounded by a joint relation with task parameters, we added task parameters (reward and force category) as coregressors, along with RT. The effect of RT was still significant for the same 10/92 neurons (3 negative and 7 positive), and it remained consistently positive across the population (t(91) = 2.38; $p = 0.020$; Fig 4K). Thus, LC neurons encoded RT just prior action onset, over and above task parameters. Note that the effect of force category was still significant for 6/92 neurons (5 positive and 1 negative) and was consistently positive across the population (t(91) = 2.86; $p = 0.0052$; Fig 4K). The effect of reward was still significant for 5/92 neurons (4 negative and 1 positive), and second-order statistics confirmed that it remained significant at the population level, with a significant negative effect (t(91) = −1.88; $p = 0.048$; Fig 4K).

We next examined the relation between LC activity before action onset and the amount of force exerted during action itself, using a GLM with only the maximum exerted force as parameter (besides the constant term). The effect was significant for 14/92 neurons (2 negative and 12 positive) and marginally significant at the population level (second-order $t$ test: t(91) = 1.68; $p = 0.096$; Fig 4L). When force category and reward were included as coregressors in the model, the influence of the exerted force on firing rate remained significant for 10/92 neurons (5 positive and 5 negative), but it showed no significant effect at the population level (second-order $t$ test: $p = 0.95$; Fig 4M). Thus, the influence of the amount of force exerted on the grip during the action on the activity of LC neurons just before action onset was negligible, compared to that of RT.

Finally, we compared the fit of the GLMs for spike count including RT, the exerted force, task parameters, and combinations of those. To run this model comparison on the entire population of neurons, as we did for the delay discounting task, we pooled all the trials of all the sessions into a single database. Then we fit 5 GLMs on the data: 1 with only RT as parameter, 1 with reward and force category as parameters, and 1 with RT, force category, and reward, 1 with the maximum exerted force alone, and 1 with force category, reward, and the maximum exerted force. The BIC of the model with RT alone was lowest (BIC = 54,960), compared to the BIC of the model with only task parameters (BIC = 54,972), with RT and task parameters (BIC = 54,969), with only the peak of force (BIC = 54,976), and with the peak of force and task parameters (BIC = 54,983). Since the difference in BIC between the best model (RT alone) and the others was greater than 5, we considered it to be significantly better than the alternative models. Thus, the activity of LC neurons before action onset in this task was more strongly influenced by RT than by the amount of force about to be exerted on the grip.

**Target detection task.** In the delay discounting and force discounting tasks, we found a positive encoding of RT prior to action onset. In both of these tasks, behavioral analyses indicates that triggering the action involved distinct levels of cognitive difficulty (and, potentially, cognitive effort), and trial-by-trial RT could be taken as a proxy for subjective difficulty and corresponding effort. But since RT modulations across conditions also included sensory-

motor constraints, especially in the force discounting task, we examined LC activity in the target detection task as a control for these effects. Indeed, in the target detection task, we could not detect any evidence for a systematic modulation of subjective difficulty/cognitive effort across conditions, but RTs showed clear modulations that could be interpreted in terms of sensory-motor constraints (see above "Behavior"). To examine modulation of LC activity as a function of RT in this task where RT only captured sensory-motor constraints, but no cognitive constraints, we fit a GLM for spike count in the [−250; 0 ms] epoch before action onset, with only RT as parameter. Only 1/26 neuron exhibited a significant (positive) encoding of RT. Moreover, the influence of RT on spike count was not significant at the population level ($p = 0.56$; Fig 4N). Since the sample of LC neurons in these experiments was clearly smaller than for the discounting experiments, where there was a significant relation between RT and LC activity before action onset, we ran a control analysis to facilitate the comparison. For both delay and force discounting tasks, we randomly selected a sample of 26 LC units and repeated the same second level analysis to evaluate the relation between firing rates and RT. In line with the results found with the whole population, there was a significant positive relation between RT and LC activity before action onset in both discounting tasks ($p < 0.05$), even when the sample of the population was the same size as in the control target detection task, where we found no relation between RT and LC activity.

Thus, in this task where the behavioral response was, modulated by sensory-motor processes, LC neurons showed no significant modulation of action triggering–related activity across task conditions and no relation with the RT.

## LC and physical effort: Force exertion–related activity

To assess the potential relation between LC and physical effort, we quantified the modulation of LC activity while monkeys were exerting a physical force on the grip. We compared LC activity across conditions both in the force discounting task (where the force was imposed and represented an effort, as measured using WTW) and in the target detection task (where we could not find any evidence of systematic differences in physical effort, since WTW was equivalent across conditions). We also examined the relation between firing rates and force production, with the hypothesis that the amount of force produced only reflected effort in the force discounting task, but not in the target detection task.

Force exertion usually lasted for about 800 ms in the force discounting task (S1A and S1B Fig), and 500 ms in the target detection task (Fig 2G), with the shortest presses lasting for about 600 ms and 400 ms, respectively. In order to avoid any overlap with the reward delivery period, we restricted the analyses to a [0; 600 ms] post action onset epoch in the force discounting task and a [0; 400 ms] epoch in the target detection task.

**Force discounting task.** We first examined the modulation of LC activity by task parameters by fitting a GLM for spike count after action onset (during force exertion) with reward and force category. The effect of reward was only significant for 11/92 neurons (3 negative and 8 positive), which is less than the numbers of neurons expected by chance ($n = 13$) given the size of that population ($n = 92$). In addition, the influence of reward was not significant across the population (second-order $t$ test, $p = 0.7$; Fig 5A). The effect of the force category was significant for 38/92 neurons (1 negative and 37 positive, example unit in Fig 4G) and was consistently positive across the population ($t(91) = 9.43$; $p < 10^{-14}$; Fig 5A).

We then examined the influence of physical force by fitting a GLM for spike count with the maximum exerted force as parameter. The effect was significant for 38/92 neurons (2 negative and 36 positive), and it was consistently positive across the population (second-order $t$ test, $t(91) = 7.39$; $p < 10^{-10}$; Fig 5B). In addition, the influence of exerted force on LC activity during

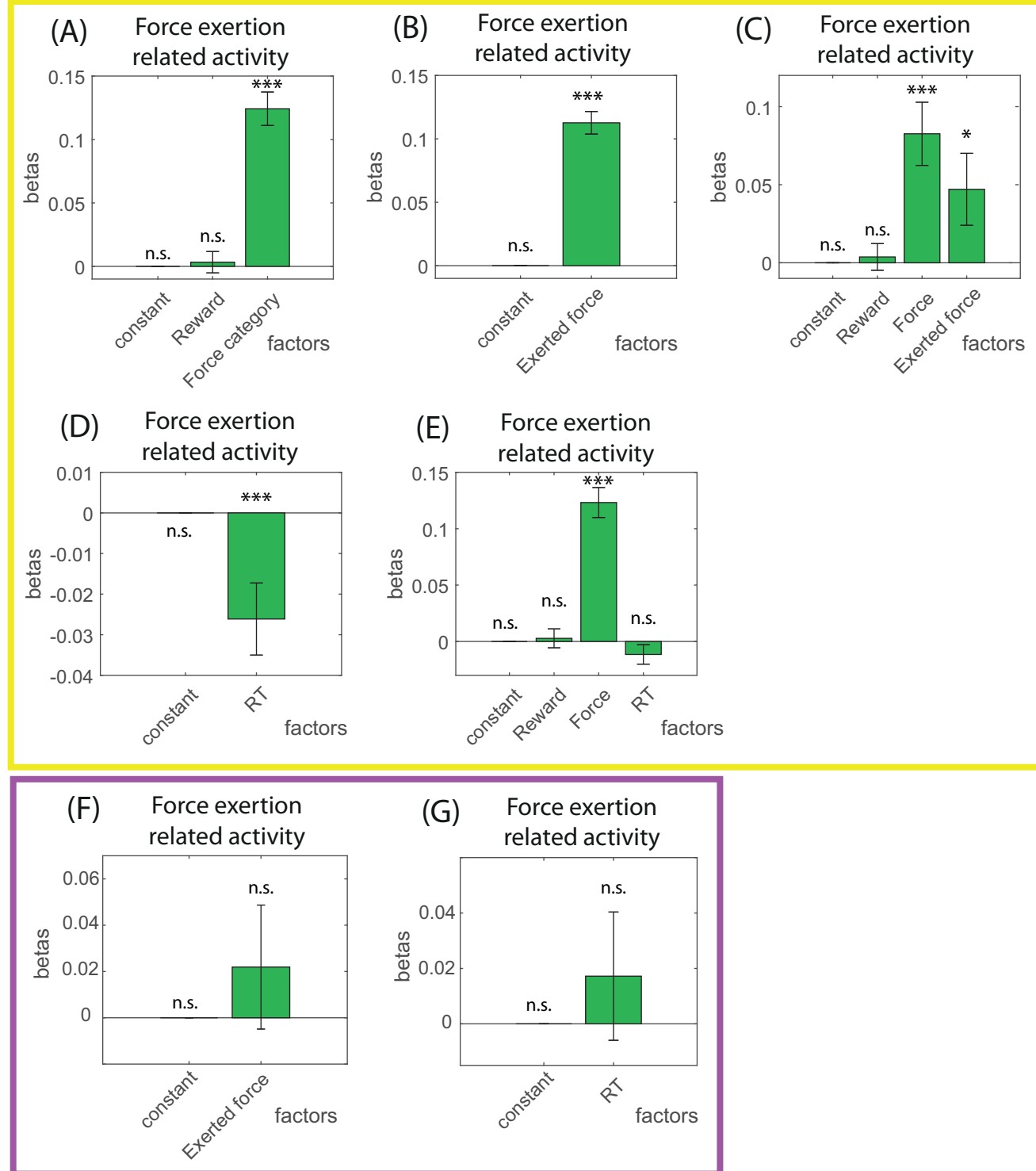

**Fig 5. Force-exertion-related activity in the force discounting and target detection tasks. (A–E)** Summary (mean and SEM) of the coefficients of the neuron-by-neuron GLMs for spike count during action (force exertion) with (A) task parameters (force category and reward) or (B) the maximum exerted force on the grip alone, (C) task parameters and the maximum exerted force, (D) RT alone, or (E) RT, reward and force category as parameters, in the force discounting task. (A) During the force exertion, LC neurons encoded the force category positively if the exerted force was not included in the model, and (B) they encoded the exerted force positively if task parameters were not included in the model. (C) In the model including task parameters and the maximum force exerted, there was a positive effect of both the exerted force and force category. (D) During the force exertion, there was a positive effect of RT on spike count if RT was alone in the model, but (E) this effect disappeared if task parameters were included in the model. (F, G) Summary (mean and

SEM) of the coefficients of the neuron-by-neuron GLMs for spike count during action (force exertion) with (F) the maximum exerted force on the grip alone, or (G) RT alone, as parameters, in the target detection task. (F) During force exertion, there was no effect of the maximum exerted force (G) or of RT in the target detection task. *: $p < 0.05$; ***: $p < 0.001$; n.s., nonsignificant. Underlying data in 10.17605/OSF.IO/PYVSA. F, force; GLM, generalized linear model; R, reward; RT, response time; SEM, standard error of the mean.

action execution remained significant when task parameters were included as coregressors (exerted force effect: still significant for 17/92 neurons, 4 negative and 13 positive; population effect: t(91) = 2.04; $p = 0.044$; Fig 5C). Note that the effect of reward was significant for 10/92 neurons (3 negative and 7 positive), but still not significant at the level of the population (second-order $t$ test: $p = 0.66$; Fig 5C). The effect of the category of force was significant for 20/92 neurons (3 negative and 17 positive), and it remained significantly positive at the population level (second-order $t$ test: t(91) = 4.08; $p < 10^{-4}$; Fig 5C). Thus, the activity of LC neurons during force production was significantly modulated by the amount of force exerted, over and above the influence of task parameters.

We then examined the influence of RT on LC activity during force exertion. The effect of RT was significant at the population level when RT was the only parameter in the GLM for spike count (significant for 10/92 neurons, 3 positive and 7 negative; second-order $t$ test: t(91) = −2.94; $p = 0.0041$). However, the effect was no longer significant when task parameters were added to the model (still significant for 10/92 neurons, second-order $t$ test: $p = 0.18$), indicating that during force exertion, LC neurons showed little sensitivity to RT.

**Target detection task.** We examined influence of the exerted force on LC activity during force exertion in the target detection task. We fit a GLM for spike count in the [0; 400 ms] postaction onset epoch with only the maximum exerted force as parameter. The effect was significant for 7/26 neurons (3 negative and 4 positive), but it was not significant across the population ($p = 0.62$; Fig 5F). Note that the effect of the exerted force was not present either before action onset (significant for 4/20 neurons, 3 positive and 1 negative, second-order $t$ test: $p = 0.56$) and that there was no effect of RT on spike count during force exertion (significant for 3/26 neurons, 2 negative and 1 positive, second-order $t$ test: $p = 0.3$; Fig 5G).

Thus, in this task where there is no evidence that force production involved a systematic difference in effort across conditions, we found no evidence for a modulation of LC activity by amount of force produced.

## Discussion

### Summary

The aim of this study was to clarify the dynamic relation between LC activity and effort production. We examined the activity of LC neurons in 2 discounting tasks (force discounting and delay discounting) and compared it to activity in a task only manipulating sensory-motor processes but not effort (target detection task). First, in both discounting tasks, triggering the behavioral response required more cognitive control in task conditions where outcome value decreased. The firing of LC neurons at the time of action-triggering scaled with task parameters in directions compatible with an encoding of the difficulty of triggering the action. In addition, trial-to-trial variability in LC activation just at action initiation was related to RT, i.e., the amount of time spent in the decision process. Second, LC neurons were activated during force production both in the force discounting task and in the control target detection task, but the magnitude of this activation only scaled with the exerted force in the force discounting task, where the amount of physical effort did vary across conditions. In the control task, where similar sensory-motor processes where involved but where we could not find any evidence of an effort contrast across conditions, LC activation was indistinguishable across conditions.

Thus, across tasks, LC neurons were activated at the time when monkeys produced an effort, either a cognitive effort to trigger a costly behavioral response or a physical effort to exert force on the grip. In addition, the magnitude of these activations scaled with the level of difficulty, and, potentially, the associated amount of effort, with a subsecond precision. Altogether, this work supports the idea that LC activity continuously monitors the production of effort.

## Behavior

Manipulating effort in animals is very challenging, since it is a complex notion and measures remain very indirect. Operationally, effort production can generally be defined as a mobilization of resources to boost performance in the face of a difficulty and meet task goals [2,7,17]. Therefore, effort can only be evaluated indirectly, through the relative modulations of performance relative to task difficulty. Here, we manipulated 2 types of difficulty: overcoming a natural tendency to disengage from the task when the expected value was low (cognitive effort) and exerting a relatively high level of force on the grip (physical effort).

For cognitive effort, we took advantage of the fact that monkeys tend to disengage from single-option tasks when the outcome value is low, even if erroneous trials are repeated [3–5,39–42]. Note that a slightly different intuition would be that monkeys simply engage in the task as a function of expected outcome value, such that in low value conditions, they fail to engage and/or wait for a better option. In such a case, the absence of engagement would be passive, in contrast with our initial intuition where the disengagement is active. But in both cases, triggering the response in correctly performed trials would require cognitive control in low value trials because it would go against a natural tendency (to actively disengage or to passively not engage). Indeed, since erroneous trials are repeated, failing to engage in a low-value trial is counterproductive, since it only increases the delay until the next reward: From an instrumental perspective, monkeys should engage in every single trial until their motivation level is low enough for them to stop performing the task altogether. Conversely, in versions of these tasks where aborted trials were not repeated, monkeys rapidly learned to skip trials associated with low value and/or high costs [43]. Thus, rhesus monkeys show a general tendency to abort (or avoid) trials associated with low benefits and/or high costs, which implies a greater difficulty to overcome this automatic response and perform the task when option value is relatively low. But since monkeys engage in a majority of trials in tasks where erroneous trials are repeated and skipping trials is not instrumental, they must be exerting cognitive control to overcome their natural tendency to abort trials associated with low reward and/or high costs, as classically described in humans and other animals performing standard discounting tasks [7,44–46]. Based on that principle, we assumed that the proportion of trials in which monkeys failed to engage in a specific task condition was a good proxy of the effort required to perform the task in that condition (subjective difficulty). In both the delay discounting and force discounting tasks, the subjective difficulty to trigger the action (and presumably the associated level of cognitive effort) decreased with expected benefits (reward size) and increased with expected costs (force or delay). By contrast, we failed to detect any evidence of a difference in engagement rate across conditions in the control target detection task, which implies that in that task the levels of subjective difficulty and associated cognitive control invested to engage in the task were relatively equivalent across conditions. Note that the fixation requirement in the force discounting task enabled us to clearly identify the engagement and disengagement of the animals, even if it might have added a significant amount of cognitive effort to the task. However, since this potential additional effort was equivalent across all conditions, it is not possible to evaluate its influence, in contrast with other factors which varied systematically across task conditions.

In addition, we used RTs as a proxy for trial-by-trial variability in subjective difficulty to trigger the action, and, potentially, to the corresponding amount of cognitive effort expensed. As classically described in the literature, RTs were affected both by decision-related processes and by sensory-motor constraints [33,47–53]. In both tasks involving force production (force discounting and target detection), RTs were strongly modulated by motor processes, as indicated by the significant relation with the amount of exerted force. But in the delay discounting task, in which sensory-motor constraints were equivalent across conditions, the modulations of RTs were in line with an interpretation in terms of subjective difficulty to trigger the action (to overcome a natural tendency to abort trials when value was low). Since in all 3 tasks monkeys remained engaged in the task from cue onset to outcome delivery (especially in the force discounting task in which gaze fixation was imposed), variations of RTs across trials were equivalent to measures of time on task. Thus, RT could provide a reliable measure of the amount of cognitive resources needed for the decision process and therefore a proxy for cognitive difficulty. Given the strong relation between difficulty and effort in those conditions, it is tempting to interpret RT variations in terms of effort, but given the numerous processes influencing RT, additional measures would be necessary to draw strong conclusions in terms of cognitive effort.

In humans studies, pupil dilation often appears to constitute a better measure of cognitive effort, compared to RT [24,54–59]. But given the complex and slow dynamics of pupil dilation it would have been difficult to use it here, especially given the strong relation between pupil and physical effort [5,60]. From a theoretical point of view, the strong correlation between an autonomic measures and effort (cognitive and physical) suggests that the central nervous system could be using a common mechanism to mobilize the resources in the face of cognitive and physical challenges. In both cases, the activation of the sympathetic system would not only increase pupil dilation but also heart rate and blood pressure, as well as the availability of metabolic resources. From a practical point of view, this also implies that pupil is not a specific measure of cognitive effort. Finally, given the strong relation between pupil dilation and LC activity, we believe that studies such as this one are critical to directly evaluate the actual relation between LC activity and potential behavioral proxies of effort, such that autonomic processes could be evaluated separately [5,23,61].

## LC activity

In both our discounting tasks, LC neurons activated just prior to action onset. As previously described, such activation was never seen for actions executed outside of tasks (between trials for example), indicating that it was probably associated to the decision process of triggering actions rather than movements per se [31,38]. In coherence with that, pupil diameter has been repeatedly shown to increase during decisions [62–66], even in the context of covert decisions [67]. Noticeably, LC neurons seem to only respond to decisions to trigger actions and not to withhold them [68], although this contrast could merely emerge from the difficulty to identify precisely the timing of decisions that are not materialized by overt actions. Indeed, since the activations of LC neurons usually consist in a few spikes (sometimes even just one), they could easily be missed given the difficultly to identify the timing of the decision to inhibit a prepotent response. Further studies could help clarifying that point.

In both the force and delay discounting tasks, the coherent but opposite modulations of LC activity and engagement rates by task parameters were in line with the idea that LC activation increases in task conditions associated with a larger amount of cognitive control to perform the task (i.e., to overcome a natural tendency to skip low value conditions). But our data indicate that beyond a general relation with task conditions, the activation of LC neurons right

before the decision could be related to the trial-by-trial modulation of cognitive difficulty and/ or effort to trigger the action. Indeed, LC neurons reliably encoded RT in that same period across the 2 discounting tasks, and model comparison confirmed that RTs predicted trial-to-trial modulation of LC activity better than task conditions alone. Critically, since LC activity did not scale with RT in the control target detection task, where RTs where only affected by sensory-motor effects, the relation between LC and RTs in both delay and force discounting tasks could reliably be interpreted in terms of cognitive difficulty and/or cognitive effort. Even if the sampled population was smaller for the control task than for the discounting tasks, where task-related modulations were significant, the fact that the effects of interest were still observed on randomly selected small samples of neurons recorded in the discounting tasks reinforces the idea that those effects are more likely to reflect effort production than sensory-motor processes.

Similarly, during force production in the force discounting task (lasting a few hundred milliseconds), LC neurons encoded the force exerted on the grip. Importantly, it was not the case in the target detection task, where exerted force had no impact on the willingness to perform the task. Thus, the activation of LC neurons during force production in the force discounting task is more likely associated with the amount of effort necessary to produce the action rather than the force itself. This interpretation is in line with previous studies in humans showing that pupil dilation, which can show strong correlations with LC activity, is associated with the experience of effort production [60]. Here, we show that the activation of LC neurons is more correlated with the subjective difficulty of exerting the force (and, potentially, with the corresponding effort produced) than with the amount of force itself.

The timing of the modulations of LC activity by subjective difficulty and possibly effort production reported here is striking, as we found that neurons encoded RT very late in the action-triggering process (at its very end, around 200 ms before action onset) and the exerted force during the execution of the action. Thus, the activation of LC neurons seems to occur after, rather than before effort production. In addition, LC axons have a very slow conduction, and action potentials were estimated to reach target cortical areas in about 130 ms [69]. Thus, LC activation is more likely to affect effort after it has been initiated. As proposed earlier, the potential influences of LC activation include both promoting ongoing functions (such as producing and monitoring the action) and promoting adaptation [1,70,71]. From an anatomical point of view, this also raises the question of which structure could provide effort related inputs to the LC. Again, effort is classically associated with various autonomic markers indicating a sympathetic activation, including pupil, heart rate, and metabolic activity [54,56,60,72–76]. Given its strong excitatory influence on the LC and its connection with the autonomic system, the nucleus Paragigantocellularis could be directly responsible for the effort-related activation of the LC (PGi, [77–81]). This might also explain why the effects of interest appear clearly at the level of the population, but not necessarily at a significant level on a large fraction of LC neurons. Indeed, since the significant effect of the population analysis implies that a majority of neurons shows a similar effect (positive or negative), even if only a fraction of those shows a significant effect at the individual level. Such a pattern might be expected, given the widespread influence of the PGi on LC neurons [77–81]. Note that in that frame, the activation of LC neurons would be just one of the numerous elements of a global autonomic activation, like all the brain regions associated with the autonomic system. But the effort-related activation of the LC could also rely upon cortical inputs, Indeed, the medial prefrontal cortex is critical for physical effort processing [19,82–85]. Indeed, a recent proposal by Silvetti and colleagues implies that the LC could mediate the boosting influence of the medial PFC on behavior [86]. Given the complexity of the relation across these structures, further studies would be necessary to address these issues.

The present study indicates that the activation of LC encodes difficulty once the animal has engaged in process of overcoming that difficulty, which we interpret in terms of effort production. Critically, this implies that the activation of LC follows the allocation of effort, such that it would be critical for effort production itself, rather than for the decision regarding how much effort should be allocated. Note that this is also in line with recent studies suggesting that, in similar tasks, the activation of LC neurons at cue onset is related to the processing of information provided by the cues, which might be interpreted in terms of mobilization of resources [3,36]. More importantly, this proposed role for LC neurons in effort production and mobilization of resources is perfectly in line with our recent pharmacological studies. Indeed, in those studies, decreasing LC activity with clonidine induced both an impairment in effort production and an increase in effort sensitivity for choices [2,4]. In other words, monkeys became more sensitive to effort when LC input was artificially decreased, in line with the idea that LC activation mediates the influence of effort allocation. If LC activation had been critical for deciding how much effort to allocate, one would have expected clonidine treatment to induce a decrease in effort sensitivity (a flattening of the relation between difficulty and behavior), as can be seen for reward sensitivity after dopamine depletions [87,88]. Besides this work emphasizing the key role of the LC/NA system in physical effort, several studies have demonstrated its critical role in various form cognitive effort including executive control and attention [21,89–98]. Thus, our data are in line with an abundant literature showing the potential implication of the LC/NA system both in physical and cognitive effort. Critically, this is (to our knowledge) the first study to identify the physiological mechanism underlying that function and proposing a unifying role accounting for the implication of LC neurons in facing both physical and cognitive challenges.

Altogether, our work refines the dynamic relation between LC activity and effort production and shows that LC neurons are activated when monkeys need to make an effort to face either a cognitive or a physical challenge. This provide a critical insight into the specific role of the LC in effort processing, but further studies would be necessary to understand how the activation of LC articulates with other key brain structures such as the cingulate cortex.

## Supporting information

**S1 Fig.** Force exertion in the force discounting task across time after action onset by **(A)** Monkey D and **(B)** Monkey A, by force categories. The thin lines represent the mean and SEM of the exerted force at each time point.
(EPS)

## Author Contributions

**Conceptualization:** Pauline Bornert, Sebastien Bouret.

**Data curation:** Sebastien Bouret.

**Formal analysis:** Pauline Bornert, Sebastien Bouret.

**Funding acquisition:** Sebastien Bouret.

**Investigation:** Pauline Bornert, Sebastien Bouret.

**Methodology:** Pauline Bornert, Sebastien Bouret.

**Project administration:** Sebastien Bouret.

**Resources:** Sebastien Bouret.

**Software:** Pauline Bornert, Sebastien Bouret.

**Supervision:** Sebastien Bouret.

**Validation:** Sebastien Bouret.

**Visualization:** Pauline Bornert, Sebastien Bouret.

**Writing – original draft:** Pauline Bornert, Sebastien Bouret.

**Writing – review & editing:** Pauline Bornert, Sebastien Bouret.

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
