## [Editor Report · Decision Letter 0]

24 Aug 2021

Dear Dr Bouret, 

Thank you for submitting your manuscript entitled "Locus coeruleus neurons encode the subjective difficulty of triggering and executing actions: evidence for a role in effort processing" for consideration as a Research Article by PLOS Biology.

Your manuscript has now been evaluated by the PLOS Biology editorial staff, and I am writing to let you know that we would like to send your submission out for external peer review.

Please re-submit your manuscript within two working days, i.e. by Aug 26 2021 11:59PM.

Kind regards,

Gabriel Gasque

Senior Editor

PLOS Biology

ggasque@plos.org

---

## [Decision Letter · Decision Letter 1]

5 Oct 2021

Dear Sebastien,

Thank you for submitting your manuscript "Locus coeruleus neurons encode the subjective difficulty of triggering and executing actions: evidence for a role in effort processing" for consideration as a Research Article at PLOS Biology. Your manuscript has been evaluated by the PLOS Biology editors, by an Academic Editor with relevant expertise, and by two independent reviewers. A third reviewer had agreed to review your manuscript as well, but s/he is now very delayed. Therefore, we have decided to move ahead with a decision. If the third reviewer belatedly submits her/his comments, I will forward them to you.

In light of the reviews (below), we are pleased to offer you the opportunity to address the comments from the reviewers in a revised version that we anticipate should not take you very long. We will then assess your revised manuscript and your response to the reviewers' comments and we may consult the reviewers again. I have included below my signature some editorial requests as well. If you address them now, this might save time when you resubmit your study. 

We expect to receive your revised manuscript within 1 month.

**IMPORTANT - SUBMITTING YOUR REVISION**

*Resubmission Checklist*

*Published Peer Review*

*PLOS Data Policy*

*Blot and Gel Data Policy*

Sincerely,

Gabriel Gasque

Senior Editor

PLOS Biology

ggasque@plos.org

EDITORIAL REQUESTS:

1) Title: We suggest a shorter title, which we think would be more appealing to a broad readership: "Locus coeruleus neurons encode the subjective difficulty of triggering and executing actions". 

2) Ethics: Please include within your manuscript the approval numbers of the protocols approved by Regional Ethical Committee for Animal Experiment (CREEA IDF n°3) and by the NIMH Animal Care and Use Committee.

3) Data: You may be aware of the PLOS Data Policy, which requires that all data be made available without restriction: http://journals.plos.org/plosbiology/s/data-availability. For more information, please also see this editorial: http://dx.doi.org/10.1371/journal.pbio.1001797

We note that you sate in our submission system that “Data are available from the ICM Institutional Data Access / Ethics Committee for researchers who meet the criteria for access to confidential data.” PLOS recognizes that, in some instances, authors may not be able to make their underlying data set publicly available for legal or ethical reasons. However, these usually only apply to research on human subjects.

If there is an ethical/legal framework that prevents or limits data release, please make these limitations clear in the Data Availability Statement. For acceptable restrictions you can read here: https://journals.plos.org/plosbiology/s/data-availability#loc-acceptable-data-access-restrictions

Note, however, that we do not require all raw data. Rather, we ask for all individual quantitative observations that underlie the data summarized in the figures and results of your paper. For an example see here: http://www.plosbiology.org/article/info%3Adoi%2F10.1371%2Fjournal.pbio.1001908#s5

These data can be made available in one of the following forms:

Regardless of the method selected, please ensure that you provide the individual numerical values that underlie the summary data displayed in the following figure panels: Figures 2A-N, 3A-F, 4CEFIKLMNO, and 5A-G.

3.a) Please also ensure that each figure legend in your manuscript includes information on where the underlying data can be found and that your supplemental data file/s has/have a legend.

3.b) Please ensure that your Data Statement in the submission system accurately describes where your data can be found.

REVIEWS:

Reviewer #1: The study aims at revealing the relationship between the Locus Coeruleus (LC) activity and effort. The authors recorded single cells activity from the LC in 5 monkeys performing different types of tasks with different levels of reward, in which they either manipulated physical effort (force exerted on a manipulandum - Force-discounting task) or 'cognitive' effort (delay to obtain the reward - delay-discounting task). Finally, in a control task, they used a target detection task to control for sensory-motor constraints (target-detection task). They recorded 75 single units (n=52 in Monkey T and n=23 in Monkey L) in the delay-discounting task, 92 LC units (in 2 monkeys, n=63 in Monkey D and n=29 in monkey A) in the Force-discounting and 26 neurons (in one monkey, Monkey J) in the target-detection task. At the behavioral level, they found that both costs and reward affected the animals' willingness to work and their response times. At the neural level, they found that neural activity in LC before and after the go signal carried information about the physical force applied, which exceeded the task parameters. Based on these results, the authors concluded that LC activity reflects the difficulty of initiating an action depending of the context. They also provide an interesting framework in which the LC/NA system might influence effort mobilization. 

The question is very interesting and the study is elegantly designed. The results are novel especially the electrophysiological recordings from the LC which are rare. They have the potential to provide key elements to our understanding on how the brain allow us to deal with challenging situations. That said, I have a number of questions and suggestions that I think the authors could address to try and improve the manuscript.

Conceptually and given the question at the heart of the study (relation between LC activity and cognitive and physical effort), it was not entirely clear to me from the introduction why the authors focused their analysis on the action onset and action execution. They should be more clear on which grounds their hypothesis was formulated (page 4, lines 64-66:"We hypothesized that LC neurons would be activated at the time when monkeys did make an effort to face the challenge at hand, both in the cognitive domain (triggering an action that they would spontaneously avoid) and in the physical domain (producing a high level of force)." 

The monkeys were familiar with the stimuli and their associated costs and benefits and the cue onset related activity might also carry information about effort, especially given that in the present study, both cognitive and physical effort were manipulated. 

Methods. 

* Pages 11 and 12: to identify LC neurons, the authors used different criterion among which "a modulation of firing rate across states of vigilance". Could the authors be more explicit on how this was done? How were the different states of vigilance qualified and/or quantified?

* Statistical analysis. The authors should provide more details about the statistical analysis carried out in the methods. As is, most of the information are disseminated in the result section and various different statistical analysis were performed such as GLM with model comparison, ANOVAs, t tests first and second order, etc…. For instance, the material in pages 16-18 currently in the result section could introduce the variables of interest and then the authors could outline and justify their statistical pipelines for behavioral and neurophysiological data.

Specifically, page 13: "We then fit a generalized linear model (GLM) per monkey on the pooled data for each combination of regressors." can you provide more details about the terms entered in the GLM? How about the random terms (e.g. sessions for behavior and the population analysis, especially when data were not z-scored?) and potential interactions terms? 

Page 15. They tested different GLM models: "one with only RT as regressor, one with only task parameters as regressors, and one with all task parameters and RT. We used Bayesian Information Criteria (BICs) to compare the fit of the models, but similar results were obtained using Akaike's information criteria (AIC)."

Did the authors statistically compare models with the different terms? Even if BIC and AIC were the lowest, were the model actually statistically compared? were they statistically different? Also, it might be easier for the reader to first have information about the model comparison and then provide the results of the best model. 

Results and discussion.

* In the Force-discounting task, there were 3 different levels of force and it is stated that "the force had been maintained above threshold". Shouldn't this also have an upper limit threshold (looks like it from fig S1)? This could be added in the methods. In this same task (but not in the others), the monkeys had to maintain fixation, which adds 'effort mobilization' to the task? How do they authors think this contributed to the neural activity in the LC? It might be worth discussing this issue. 

* The results at the population level nicely support a relation between LC activity and subjective effort in both cognitive and physical domains. Yet, when considered individually, few neurons were significantly modulated regardless of the task. Do the authors think this tells us something about the coding within LC neurons? How so? I think this disserves some discussion. 

* I am also wondering whether the direction of the effects (increase or decrease of activity) were informative more so at the action onset since this seems to be more related to the type of action during action execution? 

* I suspect that the subjective effort varied across the sessions and I understand that this might not be evident to assess these subtle changes on top of the other effects but could the authors detect such evidence looking at the effect across time and actually in any time window?

* The authors found that "The activation before action onset was especially pronounced in discounting tasks, where triggering the action could involve some form of cognitive effort. (Page 25)." Is that possible that the lack of effect in the target detection task at the population level might be related to lack of statistical power (fewer neurons sampled from only one animal)? 

* Page 20; line 383, please report beta coefficients.

* Page 26, please indicate time window

Reviewer #2: This is a fascinating paper by an established team of primate electrophysiology researchers. They have recorded in the locus coeruleus, and link the responses of neurons there to production of effort. Their analyses are focused on the question of whether firing rates in these neurons covaries with effort. They make use of a well designed set of three tasks, and on careful analysis of behavior, to establish the variables they are interested in.

The chief appeal of this paper is its focus on the LC, which is difficult to record from. As a result data from this region are lacking in the literature. As such, this report is likely to have a good deal of influence. And LC is not simply an obscure region - it has a great deal of importance for a great deal.

One of the strengths of this paper is the care that the authors put in to thinking about effort - they use a WTW measure, a reaction time measure, and a behavior measure. This, combined with careful line of thinking about how each of these variables interacts, combine to give them a good deal of insight into the deployment of effort. 

I would consider the following comments to be minor ones:

"But still, monkeys failed to initiate the action more often in trials associated with higher cost and/or lower rewards, which indicates that they could not repress a natural tendency to disengage in such conditions, even if in that situation if was counterproductive." This seems like suspect logic to me. I agree this is suggestive, but there would seem to be other explanations for these effects. It could also be there is no natural tendency, but instead monkeys engage in a natural 

The authors seem to be supporting a model in which the cost of various cognitive / physical behaviors, in terms of effort, are constant, and willingness to pay is what varies. It seems to me that the opposite is also possible - the subjective cost may vary and the willingness to expend effort may stay the same. Or alternatively, both may vary and we dont known which is which. 

"Thus, we assumed that in task conditions associated with lower willingness to work, engaging in the task and performing the action required a higher level of cognitive control to overcome the stronger tendency to disengage, compared to conditions in which average willingness to work was higher." What if the reverse is true - subjects wait until the cost is lower to do it? Then LC would mean something else.

---

## [Editor Report · Decision Letter 2]

15 Nov 2021

Dear Sebastien,

Thank you for submitting your revised Research Article entitled "Locus coeruleus neurons encode the subjective difficulty of triggering and executing actions" for publication in PLOS Biology. I have now discussed your new version with the Academic Editor, and I am pleased to tell you that we will probably accept this manuscript for publication, provided you satisfactorily address the following data and other policy-related requests:

1) Please include a README file in the OSF repository to explain how the uploaded data were analyzed to generate the plots and graphs displayed in the following figures: Figures 2A-N, 3A-F, 4CEFIKLMNO, and 5A-G

2) Please also ensure that each figure legend in your manuscript includes information on where the underlying data can be found. For example, you can write, "Underlying data in 10.17605/OSF.IO/PYVSA."

We expect to receive your revised manuscript within two weeks. 

*Published Peer Review History*

*Early Version*

Sincerely,

Gabriel Gasque, Ph.D.,

Senior Editor,

ggasque@plos.org,

PLOS Biology

---

## [Editor Report · Decision Letter 3]

17 Nov 2021

Dear Sebastien,

On behalf of my colleagues and the Academic Editor, Matthew Rushworth, I am pleased to say that we can in principle accept your Research Article "Locus coeruleus neurons encode the subjective difficulty of triggering and executing actions" for publication in PLOS Biology, provided you address any remaining formatting and reporting issues. These will be detailed in an email that will follow this letter and that you will usually receive within 2-3 business days, during which time no action is required from you. Please note that we will not be able to formally accept your manuscript and schedule it for publication until you have any requested changes.

PRESS

Sincerely, 

Gabriel Gasque, Ph.D. 

Senior Editor 

PLOS Biology

ggasque@plos.org